# A Technical-Economic Comparison between Conventional Tillage and Conservative Techniques in Paddy-Rice Production Practice in Northern Italy

**Aldo Calcante *** and **Roberto Oberti**

Department of Agricultural and Environmental Sciences, Università degli Studi di Milano, Via Celoria 2,
IT-20133 Milan, Italy; roberto.oberti@unimi.it
**\*** Correspondence: aldo.calcante@unimi.it; Tel.: +39-02-503-16465

**Abstract:** In this study a technical-economic comparison was conducted to compare three different agronomic practices applied to paddy rice cultivation areas in Italy: one based on conventional tillage (CT), and two adopting conservative agriculture approaches, namely minimum tillage (MT) and no-tillage (NT). Data about production inputs (seed, fertilizers, agrochemicals, fuel) and working time were measured for each technique during the whole production season in three experimental fields. The total production costs were computed by adding the mechanization costs, calculated through the ASABE (American Society of Agricultural and Biological Engineers) EP (Engineering Practice) 496.3 methodology, and the production input costs. The results of the study highlighted a significant reduction of total costs obtained with both minimum (−16%) and no-tillage (−19%) compared to conventional tillage.

**Keywords:** conservation agriculture; minimum tillage; no-tillage

---

## 1. Introduction

Conservation agriculture is a farming system that aims at reducing soil erosion due to intense rainfall and wind phenomena, by promoting the maintenance of a permanent soil cover, minimum soil disturbance, and diversification of plant species. It enhances biodiversity and natural biological processes above and below the ground surface, which contribute to increased water and nutrient use efficiency and to improved and sustained crop production [1].

The practice of conservation agriculture can be summarized by three pillars [2–5]:

(1) reduced soil disturbance by minimizing the mechanized operations and by avoiding inversion tillage (i.e., minimum or no-tillage);

(2) permanent organic cover of soil by crop residues and/or by cover crops between one main crop cycle and the next;

(3) crop rotation and diversification of plant species through varied crop cultivation sequences and/or associations involving at least three different crops.

Major benefits of conservation agriculture practice are also linked to maintaining soil fertility by reducing loss of organic matter and improving structure [6], as well as lowering the release of $CO_2$ in the atmosphere by enabling the accumulation of carbon in undisturbed soil with carbon sink effect [7–10], and thanks to less use of fossil fuels during tillage [11,12].

Conservation agriculture practice is often associated with the adoption of cover crops, providing specific agronomic advantages, such as improving some physic-chemical properties of the soil and

biodiversity [13], in addition to ensuring an adequate protecting cover of the soil until a new crop is grown, with additional decompaction effects and help in controlling soil-borne diseases [14,15].

Significant economic benefits for the farm are also expected from lower production cost, thanks to the reduction of the intensity of mechanized operations with savings of fuel and labor. Indeed, recent studies showed that in specific conditions, the adoption of conservation agriculture can reduce the mechanization costs up to more than 50% in the case of no-tillage farming of corn, and by even more than 75% for common wheat, due to the reduced fuel consumption and the contextual decrease of the work time which results in labor cost savings [6,8,16,17].

However, these savings do no always translate into a greater margin for the farmer, since this depends on the obtained production yield. Contrasting conclusions about the effects of conservation agriculture can be found in the scientific literature. Primarily, results on crop production seem to vary depending on the considered crop, soil, and climate [18]. For example, a multi-year research conducted in the United Kingdom on corn demonstrated that the adoption of minimum tillage techniques increased the gross margin about 6.6 % by reducing the production costs and keeping the yield unchanged [19]. Benefits in crop yield are more evident for cultivation in non-irrigated areas and/or in semi-arid conditions. In these cases, the adoption of conservative practices was associated with increasing soil water holding capacity, leading to higher crop production compared to conventional techniques [20,21].

In contrast, a multi-year research carried out in Italy showed that some crops (corn and wheat, in particular) exhibited a dramatic yield reduction of about 20% after the adoption of minimum tillage and no-tillage [8,22], while for other crops (e.g., soybean) no significant differences were found in comparison with conventional practices [22]. The authors related the reduction to a difficult weed control, and to soil compaction generated by mechanized operations carried out with non-optimal soil moisture conditions.

Concerning paddy rice cultivation, previous research showed that the adoption of minimum tillage did not affect crop yield compared to conventional tillage [8,23,24], while with no-tillage a yield decrease between 10% to 20% was observed [8,23–25]. Again, this was related to difficulty in controlling weeds (in particular *Echinochloa crus-galli* and *Oryza sativa* (L.) var. sylvatica), because of their capability to germinate and grow for extended periods of time in anaerobic conditions.

Nevertheless, given the inherent agronomic and environmental advantages, conservation agriculture was evaluated to be eligible for public subsidies in order to compensate farmers for the possible reduction in production [6,8,22,26].

The scientific literature lacks studies related to mechanization cost analysis for conservation agriculture practice, and typically they are limited to energy and labor costs. For this, the goal of the present study was to experimentally evaluate and analyze the details of total mechanization cost, including ownership and operating costs, of conventional tillage, minimum tillage, and no-tillage techniques applied to paddy rice cultivation in a typical rice farm in Northern Italy, located in a major rice producing district of Europe.

## 2. Materials and Methods

A field study was conducted in a paddy rice farm located in Pavia province (45° N, 9° E), Italy, one of the main areas of rice production in Europe, characterized by a loam soil (sand: 11%, silt: 52%, clay: 11%). Particle-size classes were determined by dispersing soil samples with Na hexametaphosphate and subsequently by applying the pipette method according to Italian standard methods for chemical analyses of soil [27]. The main soil characteristics of the field were: pH = 7.1, total nitrogen = 2.08 g/kg, phosphorus = 52.6 ppm $P_2O_5$, potassium = 173.7 ppm $K_2O$, organic matter content = 3.44%, CEC (Cation Exchange Capacity) = 17.9 meq/100 g, exchangeable calcium = 3124.3 ppm $Ca^{2+}$, exchangeable magnesium = 208.3 ppm $Mg^{2+}$, and exchangeable sodium = 40.3 ppm $Na^{2+}$.

In order to analyze the total mechanization cost associated with conventional, minimum, and no-tillage agronomic practices, three adjacent plots with the same size and shape were obtained from a whole field (Figure 1).

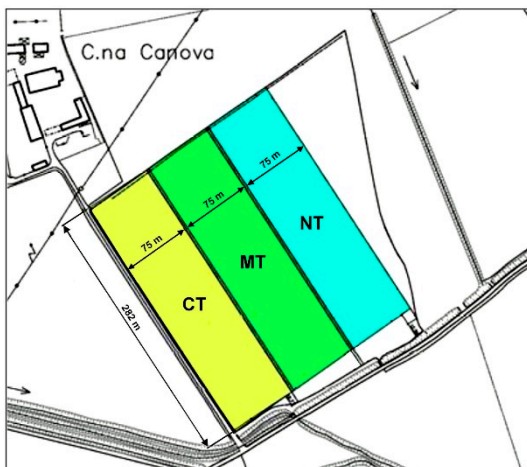

**Figure 1.** The three experimental plots used for the field study (CT: conventional tillage, MT: minimum tillage, NT: no-tillage).

The three plots were each 2.11 ha in size, and were delimited by means of a topographic survey, and traced with centimeter accuracy by a tractor equipped with a rotary ditcher and an RTK (Real Time Kinematic) automatic guidance system. In the previous three years, the field was cultivated according to minimum tillage practice (non-inversion tillage with working depth of 10 cm) and was regularly sown with cover crops between two main crops. After the last harvest of soybean (October 2017), the field was sown with a cover crop mixture of vetch (*Vicia sativa* L.) and rapeseed (*Brassica napus* L.).

For the experimental tests, rice (*Oryza Sativa* L.) ssp japonica, cultivar *Caravaggio* was sown. This commercial cultivar has a 147-day nominal growth cycle for the farm area, a significant resistance to main fungal diseases (e.g., *Magnaporthe oryzae*), and high yield potential.

The three plots, named CT (conventional tillage), MT (minimum tillage), and NT (no-tillage), respectively, underwent different soil preparation to seeding. The other crop operations—fertilization (3 distributions), pre-emergence (1 application) and post-emergence (1 application) herbicide treatments, fungicide treatments (2 applications), and harvesting—were exactly the same (same dates, same amount, and dose).

Finally, in order to compare fuel consumption and work time associated with the agronomical practices, the same tractors were used in three plots (tractor 1: 144 kW rated engine power; tractor 2: 97 kW rated engine power; both four-wheel drive (4WD)), as it was the same combine harvester used for the harvest.

The soil preparation in the three plots according to the three different practices is detailed as follows.

### 2.1. Conventional Tillage (CT)

The CT practice started with ploughing with a 4-share reversible plough, coupled to tractor 1. The plough working depth was set at 27 cm, enough for the complete burying of the autumn cover crop. Subsequently, a pre-sowing fertilization with urea (N 46) was carried out by using a centrifugal fertilizer spreader, coupled to tractor 2, followed by one harrowing pass with a working depth of 10 cm to smoot the plot surface. After the secondary tillage, the soil was in ideal conditions to apply the false seedbed technique (i.e., a seedbed preparation earlier than the real sowing date with the purpose of allowing time for germination and subsequent destruction of weeds), ahead of establishing a crop in the real seedbed [16]. Once weeds emerged, a non-selective herbicide (glyphosate) was applied.

The formation of a surface crust required a second harrowing conducted by a tine harrow with 3 cm of working depth, in order to create an optimal soil condition for seeding. Sowing was carried out using a pneumatic seed drill

### 2.2. Minimum Tillage (MT)

The MT practice started with the pre-sowing fertilization of the plot with urea by a centrifugal fertilizer spreader coupled to tractor 2. The tillage operations were limited to disk harrowing with a working depth of 10 cm. Cover crop was properly destroyed, even if with the formation of coarse soil clod that required a second disk harrowing at the same depth but conducted in transverse direction compared to the previous one; this obtained an optimal seedbed both for soil refinement and residue burying. Once tillage operations were concluded, the seeding was delayed in order to allow weeds germination (in particular *Oryza sativa* (L.) var. sylvatica) and subsequent control by a non-selective herbicide treatment. Sowing was carried out using a combined seed drill.

### 2.3. No-Tillage (NT)

The NT practice began with the pre-sowing fertilization of the plot with urea by a centrifugal fertilizer spreader, followed by a chemical termination of the cover crop with glyphosate. Thereafter, in order to facilitate the sowing, the cover crop residues were lodged in the seeding direction by using a Cambridge roller. The seeding was carried out using a sod-seeder with a preset load of 2.8 kN per planter unit. The seeding rate was the same as the one adopted in CT and MT (200 kg/ha).

### 2.4. Production Factors

Table 1 shows the sequence of the mechanized operations adopted in the three plots, together with the amount of production input used (seeds, fertilizers, pesticides).

### 2.5. Field Capacity

The work time of every mechanized operation conducted was recorded by a single frequency GPS (Global Positioning System) receiver (ArvaPc, Arvatec Srl, Milan, Italy) installed on the two tractors used for field operations. The GPS recorded, with a frequency acquisition of 1 Hz, the date, time, and position of the tractor. The generated log file was in NMEA 0183 format. The recorded data were analyzed to compute for every conducted operation the effective field capacity ($C_a$, ha/h); that is the actual rate of land processed per time unit [28], calculated as follows:

$$C_a = \frac{A}{wt} \tag{1}$$

where, *A* = area processed by the equipment (ha) (2.11 ha in the case of the operation carried out on the single plot, 6.33 ha regarding operations carried out at the same time on the three plots); *wt* = total work time measured by the GPS receiver (*h*), it includes the actual operating time, turnings time, and filling time necessary to refill seed hoppers, fertilizer hopper, and sprayer's tank.

By adding the work time measured for every mechanized operation, the total work time (*h*) necessary to operate CT, MT, and NT practices in the three plots was computed.

Table 2 shows the main technical parameters of the equipment used and the couplings between tractors and the specific operating machines. Furthermore, Table 3 reports the rated power *Pn* (kW) for the two 4WD tractors and the working width *Dr* (m) for the operating machines.

**Table 1.** Sequence of the mechanized operations carried out on the three experimental plots.

| Conventional Tillage (CT) | | | Minimum Tillage (MT) | | | No-Tillage (NT) | | |
|---|---|---|---|---|---|---|---|---|
| Operation | Production Factor | Rate (kg/ha) | Operation | Production Factor | Rate (kg/ha) | Operation | Production Factor | Rate (kg/ha) |
| Ploughing | - | - | 1st Fertilization | Urea | 75 | 1st Fertilization | Urea | 75 |
| 1st Fertilization | Urea | 75 | 1st Disc Harrowing | - | - | Herbicide Treatment | Glyphosate | 2 |
| Disc Harrowing | - | - | 2nd Disc Harrowing | - | - | Rolling | - | - |
| Herbicide Treatment | Glyphosate | 2 | Herbicide Treatment | Glyphosate | 2 | Sod Seeding | Caravaggio cv. Seed | 200 |
| Tine Harrowing | - | - | Seeding | Caravaggio cv. Seed | 200 | | | |
| Seeding | Caravaggio cv. Seed | 200 | | | | | | |
| | | | Pre-emergence Herbicide* | Pendimetalin + Clomazone | 1.8 + 0.3 | | | |
| | | | 2nd Fertilization* | Slow-release Nitrogen | 105 | | | |
| | | | Post-emergence Herbicide* | Penoxsulam + Terbuthylazine + Bromoxinil | 2 + 0.15 + 1.2 | | | |
| | | | 1st Fungicide Treatment* | Tricyclazole | 0.3 | | | |
| | | | 3rd Fertilization* | 19-0-22 Complex | 180 | | | |
| | | | 2nd Fungicide Treatment* | Tricyclazole + Azoxystrobin | 0.3 + 0.75 | | | |
| | | | Harvesting* | - | - | | | |

*: common to all the three agronomic practices.

**Table 2.** Main technical parameters of the equipment and coupled tractors used in the three plots of the study.

| Operation | Conventional Tillage (CT) | | | Minimum Tillage (MT) | | | No-Tillage (NT) | | |
|---|---|---|---|---|---|---|---|---|---|
| | 4WD Tractor | Operating Machine | | 4WD Tractor | Operating Machine | | 4WD Tractor | Operating Machine | |
| | Pn (kW) | Typology | Dr (m) | Pn (kW) | Typology | Dr (m) | Pn (kW) | Typology | Dr (m) |
| Ploughing | 144 | 4-share Plough | 1.7 | n.c. | n.c. | n.c. | n.c. | n.c. | n.c. |
| Disc Harrowing | n.c. | n.c. | n.c. | 144 | Disk Harrow | 5.0 | n.c. | n.c. | n.c. |
| Harrowing | 144 | Rotary Harrow | 5.0 | n.c. | n.c. | n.c. | n.c. | n.c. | n.c. |
| Fertilization | 97 | Fertilizer Spreader | 20.0 | 97 | Fertilizer Spreader | 20.0 | 97 | Fertilizer Spreader | 20.0 |
| Pesticides Distribution | 97 | Sprayer | 24.0 | 97 | Sprayer | 24.0 | 97 | Sprayer | 24.0 |
| Breaking Surface Crust | 144 | Tine Harrow | 5.0 | n.c. | n.c. | n.c. | n.c. | n.c. | n.c. |
| Rolling | n.c. | n.c. | n.c. | n.c. | n.c. | n.c. | 97 | Cambridge Roller | 3.0 |
| Seeding | 97 | Pneumatic Seed Drill | 4.5 | 97 | Combined Seed Drill | 4.5 | 97 | No-till Drill | 3.0 |

Abbreviations: n.c.: mechanized operation not carried out in the specific agronomic practice; Dr: working width of the various operating machines; 4WD: four-wheel drive.

*2.6. Mechanization Costs Calculation*

In order to evaluate the possible profitability of MT and NT in comparison to CT, the total costs associated with the use of each typology of equipment were computed applying the methodology using the ASABE EP 496.3 methodology [28]. This is a reference method for accounting agricultural machinery costs by evaluating their annual ownership costs (€/year) and operating costs (€/h) [29,30]. Ownership is independent of machine use, while operating costs are proportional to the utilization of the machine. Total machine costs are the sum of the ownership and operating costs [28]. In particular, ownership costs include equipment depreciation, interest on the investment, taxes, insurance, and housing of the machine [31].

Depreciation is the reduction in the value of a machine with time and use. It is often the largest single cost of machine ownership and considers the salvage value of the machine at the end of its life.

The cost of ownership includes the interest on the money that is invested in the machine. Typically, a loan is used to purchase the machine; in this case the interest rate is known. If a machine is purchased for cash, the relevant interest rate is the rate that could have been obtained if the money had been invested instead of being used to purchase the machine.

Taxes include sales tax assessed on the purchase price of a machine and property tax assessed on the remaining value in any given year. Insurance is usually related to the civil liability in case of an accident. The cost for housing takes into account the investment for the shelter to recover the agricultural machine. The annual cost of shelter is considered to be constant over the life of the machine.

The ownership costs $C_o$ (€/yr) were calculated through the following equation [28]:

$$C_o = P \times \frac{1 - (1 - t_d)^{L+1}}{L} + \frac{1 + (1 - t_d)^{L+1}}{2} \times i + K_2 \qquad (2)$$

where, $P$ = purchase price of the machine (€); $t_d$ = depreciation rate of machine (%); $L$ = machine life (yr); $i$ = annual interest rate (%); $K_2$ = ownership cost factor for taxes, housing, and insurance (usually 1.5 % of $P$)

Operating costs are the costs associated with use of a machine and include the costs of fuel and oil, repair and maintenance, and labor.

The cost of fuel for the tractor/combine involved was calculated by measuring the actual fuel consumption during each plot operation (see Equation (6)), multiplied by the market price of fuel.

The cost of lubricant oil was calculated by multiplying the market price of oil by the hourly oil consumption ($Q_i$, kg/h) calculated by the following equation [32]:

$$Q_i \ = \ \rho_{oil} \ \times \ (0.000239 \ \times \ P_r \ + \ 0.00989) \tag{3}$$

where, $\rho_{oil}$ = lubricant oil density (0.880 kg/ dm$^3$); $Pr$ = rated engine power (kW).

Costs for repairs and maintenance are highly variable, depending on the care provided by the farmer. Repair and maintenance cost (Crm, €/h) tend to increase with the size, complexity, and the working hours of the machine [33]:

$$C_{rm} \ = \ P \ \times \ FR \ \times \ \frac{(L \ \times \ H_a)^{RF2-1}}{(Sl)^{RF2}} \tag{4}$$

where, $P$ = purchase price of the machine (€); $FR$ = repair and maintenance factor (% of P); $L$ = machine life (yr); $H_a$ = yearly working hours of the specific machine (h/yr); $RF2$ = repair and maintenance factor; $Sl$ = estimated life of the machine (h).

All the parameters of Equations (2) and (4) are listed in Table 3.

Equation (4) (ASABE EP 496.3 [28] modified by Lazzari and Mazzetto [34]) provides the hourly repair and maintenance cost as a function of the yearly working hours of the specific machine.

Ownership, operating, and total machine costs can be calculated on an hourly, or per-ha basis. Total per-ha cost (C$_{tot}$, €/ha·yr) is calculated by dividing the total annual cost of the area covered by the machine during the year, or by the area involved in a particular mechanized activity:

$$C_{tot} \ = \ \frac{C_o \ + \ (C_{fo} \ + \ C_{rm} \ + \ C_l) \ \times \ H_a}{A} \tag{5}$$

where, $C_o$ = ownership costs (€/yr); $C_{fo}$ = costs for fuel and lubricant oil (€/h); $C_{rm}$ = repair and maintenance costs (€/h); $C_l$ = labor cost (€/h); $H_a$ = yearly working hours of the specific machine (h/yr); $A$ = considered area (ha).

Table 3 lists the economic parameters used for applying the ASABE EP 496.3 methodology [28] for every equipment.

After each operation, the volume of diesel consumed was measured by refilling the fuel tank of the tractor/harvester by using a graduated transparent container, and per-ha fuel consumption (kg/ha) was computed as:

$$\text{Fuel consumption} \ = \ \rho_{diesel} \ \times \ \frac{x}{A} \tag{6}$$

where, $\rho_{diesel}$ = diesel density (0.835 kg/dm$^3$); $x$ = volume of diesel consumed for each operation (dm$^3$); $A$ = area processed by the equipment (2.11 ha in the case of the operation carried out on the single plot, 6.33 ha for the operations carried out at the same time on the three plots).

**Table 3.** Economic parameters used for applying the ASABE EP 496.3 methodology [28] for every considered equipment.

| Agricultural Machine | Purchase Price* (€) | Depreciation Rate** (%) | Machine Life** (yr) | Estimated Life** (h) | Annual Interest Rate* (%) | FR** (%) | RF2** (-) | Labor Cost* (€/h) |
|---|---|---|---|---|---|---|---|---|
| Tractor 1 | 118,800 | 12.5 | 12 | 12000 | 3.5 | 80 | 2.0 | 20 |
| Tractor 2 | 58,000 | | | | | | | |
| 4-share Plough | 18,800 | 18 | 12 | 2000 | 3.5 | 100 | 1.8 | |
| Disk Harrow | 30,000 | 18 | 12 | 2000 | 3.5 | 60 | 1.7 | |
| Rotary Harrow | 19,800 | 19.5 | 10 | 2000 | 3.5 | 80 | 2.2 | |
| Fertilizer Spreader | 6100 | 21 | 8 | 1500 | 3.5 | 70 | 1.3 | |
| Sprayer | 40,000 | 25.5 | 6 | 2000 | 3.5 | 60 | 1.3 | - |
| Tine Harrow | 18,000 | 19.5 | 10 | 2000 | 3.5 | 70 | 1.4 | |
| Cambridge Roller | 6000 | 19.5 | 12 | 2000 | 3.5 | 70 | 1.3 | |
| Pneumatic Seed Drill | 45,600 | 21 | 8 | 1500 | 3.5 | 75 | 2.1 | |
| Combined Seed Drill | 45,600 | 21 | 8 | 1500 | 3.5 | 75 | 2.1 | |
| No-till Drill | 32,000 | 21 | 8 | 1500 | 3.5 | 75 | 2.1 | |

* Typical current values for Italian market. The labor cost is related to the tractor driver only. ** According to [35].

In order to compute the costs related to diesel and lubricant oil consumption associated to each operation, a price of 1 €/kg for diesel and 3.5 €/kg for lubricant oil was considered [32].

The results obtained for the three plots were then scaled-up to a paddy rice farm area of 75 ha. This farm size was chosen because it is typical for the producing area considered in the study, as well as because the field capacity of the machines considered would accomplish the sequence of operations in the available time for field work, without the need of additional units of equipment.

The total costs per ha was hence obtained by summing the cost of the production factors used (seed, fertilizers, agro-chemicals) and the mechanization costs (including labor cost), calculated for each considered tillage practice. The harvest of paddy rice was made by a combine contractor at a cost of 250 €/ha

## 3. Results and Discussion

Table 4 shows the dates on which the mechanized operations were carried out in the three experimental plots, and the related effective field capacity (ha/h) calculated from the GPS data recorded during the field activities. As expected, the lowest field capacity was found for conventional tillage (ploughing and harrowing, with an effective field capacity of 0.76 ha/h and 1.5 ha/h, respectively), and for seeding (from 1.2 to 1.7 ha/h), due to the small working width of the machines. On the contrary, the highest field effective capacity was found for operations conducted with large working width (i.e., fertilizations 9.3–10.5 ha/h and protection treatments 8–10.5 ha/h) for all the CT, MT, and NT plots.

**Table 4.** Measured effective field capacity ($C_a$, ha/h) of the operations carried out in the three experimental plots. All operations were carried out in 2018.

| Conventional Tillage (CT) | | | Minimum Tillage (MT) | | | No-Tillage (NT) | | |
|---|---|---|---|---|---|---|---|---|
| Date | Operation | $C_a$ (ha/h) | Date | Operation | $C_a$ (ha/h) | Date | Operation | $C_a$ (ha/h) |
| 23 April | Ploughing | 0.76 | n.c. | n.c. | n.c. | n.c. | n.c. | n.c. |
| 24 April | Harrowing | 1.5 | 24 April | 1st Fertilization | 9.3 | n.c. | n.c. | n.c. |
| 24 April | 1st Fertilization | 9.3 | 25 April | 1st Disc Harrowing | 2.7 | 24 April | 1st Fertilization | 9.3 |
| 15 May | Herbicide Treatment | 8 | 25 April | 2nd Disc Harrowing | 2.1 | 15 May | Herbicide Treatment | 8 |
| 16 May | Breaking Surface Crust | 3.6 | 15 May | Herbicide Treatment | 8 | 17th May | Rolling | 3.1 |
| 17 May | Seeding | 1.7 | 17 May | Seeding | 1.8 | 17 May | Sod Seeding | 1.2 |
| 17 May | Pre-emerge Herbicide Treatment | 13 | 17 May | Pre-emerge Herbicide Treatment | 13 | 17 May | Pre-emerge Herbicide Treatment | 13 |
| 20 June | 2nd Fertilization | 10.5 | 20 June | 2nd Fertilization | 10.5 | 20 June | 2nd Fertilization | 10.5 |
| 21 June | Post-emerge Herbicide Treatment | 13 | 21 June | Post-emerge Herbicide Treatment | 13 | 21 June | Post-emerge Herbicide Treatment | 13 |
| 14 July | 1st Fungicide Treatment | 9.9 | 14 July | 1st Fungicide Treatment | 9.9 | 14 July | 1st Fungicide Treatment | 9.9 |
| 20 July | 3rd Fertilization | 9.3 | 20 July | 3rd Fertilization | 9.3 | 20 July | 3rd Fertilization | 9.3 |
| 27 July | 2nd Fungicide Treatment | 9.9 | 27 July | 2nd Fungicide Treatment | 9.9 | 27 July | 2nd Fungicide Treatment | 9.9 |
| 17 October | Harvesting | 1.2 | 17 October | Harvesting | 1.2 | 17 October | Harvesting | 1.2 |

Abbreviation: n.c.: mechanized operation not carried out in the specific agronomic practice.

Table 5 shows the fuel consumption (kg/ha and kg/h of diesel) for every operation. Again, the highest fuel consumption was found for ploughing (34.1 kg/ha of diesel) and rotary harrowing (18.9 kg/ha), both used only in CT practice. Note that ploughing, rotary harrowing, disc harrowing, and the tine surface harrowing were carried out by the 144-kW tractor 1, whilst for the other activities the 97-kW tractor 2 was used. The fuel consumption related to the seeding was the same for CT and MT (6.9 kg/ha), while it was lower for NT (5.7 kg/ha), due to the typology of the seed drill used. Finally, fuel consumption for paddy rice harvesting was 17.3 kg/ha of diesel.

**Table 5.** Fuel consumption measured during the field operations.

| Conventional Tillage (CT) | | | Minimum Tillage (MT) | | | No-Tillage (NT) | | |
|---|---|---|---|---|---|---|---|---|
| Operation | Fuel Cons. (kg/ha) | Fuel Cons. (kg/h) | Operation | Fuel Cons. (kg/ha) | Fuel Cons. (kg/h) | Operation | Fuel Cons. (kg/ha) | Fuel Cons. (kg/h) |
| Ploughing | 34.1 | 25.9 | n.c. | n.c. | n.c. | n.c. | n.c. | n.c. |
| Harrowing | 18.9 | 28.4 | 1st Fertilization | 0.9 | 8.4 | n.c. | n.c. | n.c. |
| 1st Fertilization | 0.9 | 8.4 | 1st Disc Harrowing | 6.5 | 17.6 | 1st Fertilization | 0.9 | 8.4 |
| Herbicide Treatment | 1.3 | 11.4 | 2nd Disc Harrowing | 7 | 14.7 | Herbicide Treatment | 1.3 | 11.4 |
| Breaking Surface Crust | 4.4 | 15.8 | Herbicide Treatment | 1.3 | 11.4 | Rolling | 2.6 | 8.0 |
| Seeding | 6.9 | 11.7 | Seeding | 6.9 | 12.4 | Sod Seeding | 5.7 | 7.1 |
| Pre-emergence Herbicide treatment | 1.3 | 16.9 | Pre-emergence Herbicide Treatment | 1.3 | 16.9 | Pre-emergence Herbicide Treatment | 1.3 | 16.9 |
| 2nd Fertilization | 0.9 | 9.5 | 2nd Fertilization | 0.9 | 9.5 | 2nd Fertilization | 0.9 | 9.5 |
| Post-emergence Herbicide Treatment | 1.3 | 16.9 | Post-emergence Herbicide Treatment | 1.3 | 16.9 | Post-emergence Herbicide Treatment | 1.3 | 16.9 |
| 1st Fungicide Treatment | 1.3 | 12.9 | 1st Fungicide Treatment | 1.3 | 12.9 | 1st Fungicide Treatment | 1.3 | 12.9 |
| 3rd Fertilization | 0.9 | 8.4 | 3rd Fertilization | 0.9 | 8.4 | 3rd Fertilization | 0.9 | 8.4 |
| 2nd Fungicide Treatment | 1.3 | 12.9 | 2nd Fungicide Treatment | 1.3 | 12.9 | 2nd Fungicide Treatment | 1.3 | 12.9 |
| Harvesting | 17.3 | 20.8 | Harvesting | 17.3 | 20.8 | Harvesting | 17.3 | 20.8 |
| Total | 90.8 | - | Total | 46.9 | - | Total | 34.1 | - |

n.c.: mechanized operation not carried out in the specific agronomic practice.

Overall, the total fuel consumption for the three agronomic practices was 90.8 kg/ha for CT, 46.9 kg/ha for MT, and 34.1 kg/ha for NT, corresponding to fuel savings of 48% and 63% for MT and NT, respectively, compared to CT.

These findings are quite in accordance with those obtained by Rognoni et al. [8] for wheat cultivation in Italy, with fuel savings of 42% for MT and of 75% for NT, compared to CT. Similarly, for corn, they found 57% (MT) and 61% (NT) savings compared to CT. Studying wheat cultivation in the United Kingdom, Morris et al. [35] obtained fuel savings of 32% for MT and of 77% for NT, compared to CT, but only considering tillage without accounting for the consumption associated with other mechanized operations (fertilizing, pesticides distributions, harvesting).

By scaling up the experimental results obtained in the three plots on a rice farm area of 75 ha, the working hours required by paddy rice cultivation with CT, MT, and NT practices are shown in Table 6. In overall, the working time for CT was 335.4 h, MT was 227.0 h, and NT was 208.5 h, with work savings for MT and NT of 32% and 38%, respectively, compared to CT.

**Table 6.** Computed work times for 75 ha paddy farm with the three considered agronomic practices.

| Conventional Tillage (CT) | | Minimum Tillage (MT) | | No-Tillage (NT) | |
|---|---|---|---|---|---|
| Operation | Work Time Per 75 ha Area (h) | Operation | Work Time Per 75 ha Area (h) | Operation | Work Time Per 75 ha Area (h) |
| Ploughing | 98.7 | n.c. | n.c. | n.c. | n.c. |
| Harrowing | 50.0 | 1st Fertilization | 8.1 | n.c. | n.c. |
| 1st Fertilization | 8.1 | 1st Disc Harrowing | 27.8 | 1st Fertilization | 8.1 |
| Herbicide Treatment | 9.4 | 2nd Disc Harrowing | 35.7 | Herbicide Treatment | 9.4 |
| Breaking Surface Crust | 20.8 | Herbicide Treatment | 9.4 | Rolling | 24.2 |
| Seeding | 44.1 | Seeding | 41.7 | Sod Seeding | 62.5 |
| Pre-emergence Herbicide Treatment | 5.8 | Pre-emergence Herbicide Treatment | 5.8 | Pre-emergence Herbicide Treatment | 5.8 |
| 2nd Fertilization | 7.1 | 2nd Fertilization | 7.1 | 2nd Fertilization | 7.1 |
| Post-emergence Herbicide Treatment | 5.8 | Post-emergence Herbicide Treatment | 5.8 | Post-emergence Herbicide Treatment | 5.8 |
| 1st Fungicide Treatment | 7.6 | 1st Fungicide Treatment | 7.6 | 1st Fungicide Treatment | 7.6 |
| 3rd Fertilization | 8.1 | 3rd Fertilization | 8.1 | 3rd Fertilization | 8.1 |
| 2nd Fungicide Treatment | 7.6 | 2nd Fungicide Treatment | 7.6 | 2nd Fungicide Treatment | 7.6 |
| Harvesting | 62.3 | Harvesting | 62.3 | Harvesting | 62.3 |
| Total | 335.4 | Total | 227.0 | Total | 208.5 |

n.c.: mechanized operation not carried out in the specific agronomic practice.

The total time necessary to cultivate one hectare was 4.5 h/ha for CT, 3.0 h/ha for MT, and 2.8 h/ha for NT. Considering an hourly labor cost of 20 €/h, it follows that the labor cost per hectare is 72.8 €/ha for CT, 43.9 €/ha for MT, and 38.9 €/ha for NT. Morris et al. [35] found that the total time necessary for tillage operations on one hectare of wheat is 2.5 h/ha for CT, 1 h/ha for MT, and 0.5 h/ha for NT.

In this study, the main factors of the mechanization operating costs (fuel + labor) resulted in 163.6 €/ha for CT, 90.8 €/ha for MT, and 73.0 €/ha for NT, with savings of 46% and 55% of conservative techniques compared to CT; that was in fair agreement with [36].

By considering the total costs of mechanization for the machines used in the study and scaling up to the case of a 75 ha farm size, the differences in costs were relatively less marked (Table 7) than the comparison to the simple sum of diesel and labor costs for the three considered practices. In fact, the total costs of mechanization for a 75 ha paddy rice farm, calculated through the methodology defined in the ASABE EP 496.3 standard [28] (assuming an annual use of 500 h for both tractors) was 604.8 €/ha for CT, 424.8 €/ha for MT, and 382.7 €/ha for NT, with savings of 30% and 37%, respectively. In a study on soybean cultivation in the USA, McIsaac et al. [17] obtained savings of 16% and 27% for MT and NT, respectively, by only accounting tillage operations and without considering the incidence of production factors and other operations.

Considering that the costs per hectare for seed cv. Caravaggio, fertilizer, and herbicide and fungicide resulted 195.0 €/ha, 129.9 €/ha, and 332.2 €/ha respectively, the total costs (mechanization costs, labor cost, cost for seed, fertilizer, and agro-chemicals) related to the three agronomic practices were finally computed (Table 8). De facto, since the cost for the factors of production is the same for CT, MT, and NT, the observed differences were only due to the mechanization costs for tillage, and to the labor requirement necessary to conclude the considered agronomic practices.

**Table 7.** Total costs of mechanization by referring a paddy area of 75 ha.

| Conventional Tillage (CT) | | | Minimum Tillage (MT) | | | No-Tillage (NT) | | |
|---|---|---|---|---|---|---|---|---|
| Operation | Hourly Cost (€/h) | Cost Per Hectare (€/ha)* | Operation | Hourly Cost (€/h) | Cost Per Hectare (€/ha)* | Operation | Hourly Cost (€/h) | Cost Per Hectare (€/ha)* |
| Ploughing | 82.20 | 108.16 | n.c. | n.c. | n.c. | n.c. | n.c. | n.c. |
| Harrowing | 108.42 | 72.28 | n.c. | n.c. | n.c. | n.c. | n.c. | n.c. |
| Fertilization | 63.29 | 6.52 | Fertilization | 63.29 | 6.52 | Fertilization | 63.29 | 6.52 |
| Distribution of Agrochemicals | 233.68 | 21.24 | Disc Harrowing | 120.01 | 42.50 | Distribution of Agrochemical | 233.68 | 21.24 |
| Breaking Surface Crust | 145.46 | 40.41 | Distribution of Agrochemicals | 233.68 | 21.24 | Rolling | 53.29 | 17.19 |
| Seeding | 180.60 | 106.24 | Seeding | 188.20 | 104.56 | Sod Seeding | 105.30 | 87.75 |
| Harvesting | - | 250.00 | Harvesting | - | 250.00 | Harvesting | - | 250.00 |
| Total | - | 604.85 | Total | - | 424.82 | Total | - | 382.71 |

Abbreviations: n.c.: mechanized operation not carried out in the specific agronomic practice. * Calculated for every mechanized operation as the ratio between the hourly cost and the effective field capacity $C_a$.

**Table 8.** Total costs of paddy rice production according the three agronomic practices on a paddy area of 75 ha, and the savings achievable by MT and NT in comparison with CT.

| Costs Per Hectare (€/ha) | Conventional Tillage (CT) | Minimum Tillage (MT) | No-Tillage (NT) |
|---|---|---|---|
| Mechanization Costs | 604.85 | 424.82 | 382.71 |
| Labor Cost | 72.79 | 43.87 | 38.94 |
| Seed | 195.0 | 195.0 | 195.0 |
| Fertilizer | 129.90 | 129.90 | 129.90 |
| Pesticides | 332.2 | 332.2 | 332.2 |
| Total | 1334.74 | 1125.79 | 1078.75 |
| Saving in Comparison with CT (%) | - | 16 % | 19 % |

Finally, considering the total costs to produce paddy rice, including mechanization, labor, seed, fertilizer, and pesticides, the total cost per hectare amounted to 1334.7 €/ha for CT, 1125.8 €/ha for MT, and 1078.7 €/ha for NT, with total savings of 16% and 19%, respectively These findings demonstrate that from the production costs point of view, conservation agriculture can be more sustainable than conventional approaches. It should be recalled, however, that conservation agriculture techniques do not always allow levels of production comparable with those obtained with conventional approaches. In the case of a decrease in yield, despite the reduction of mechanization costs due to the conservation approaches, the economic balance can be uncertain for farmers.

## 4. Conclusions

This study aimed to contribute to filling the gap in the scientific literature about the technical-economic analysis of conservation agriculture approaches in paddy rice cultivation, compared with conventional practices. With a comparative experiment based on direct field measurements of data about machinery and production factors used, the analysis showed that the adoption of conservative techniques for paddy rice cultivation allowed significant savings on production costs, thanks to reduced work time (47%–61% less) and to lower mechanization costs (42%–58% less) in comparison to conventional practices. Moreover, the reduction found in fuel consumption (48%–63% less) with the associated reduction of emissions can also be related to the direct environmental benefits.

**Author Contributions:** Conceived and designed the experiments, performed the experiments, collected and analyzed the data, interpreted the results and developed the manuscript, A.C.; conceived and designed the experiments, performed the experiments, collected and analyzed the data, interpreted the results and developed the manuscript, R.O.

**Funding:** This research received no external funding.

**Acknowledgments:** The authors would like to acknowledge the Azienda Agricola Sgariboldi (Torrevecchia Pia, Pavia, Italy) for hosting the field tests.

**Conflicts of Interest:** The authors declare no conflict of interest.

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
