# Peer review of "A Technical-Economic Comparison between Conventional Tillage and Conservative Techniques in Paddy-Rice Production Practice in Northern Italy"

_agronomy, doi:10.3390/agronomy9120886_

Round 1

Reviewer 1 Report

After reviewing the corrections made, I must say that they are satisfactory.

Reviewer 2 Report

 A technical-economic comparison between conventional tillage and conservative techniques in paddy-rice production practice in northern Italy.  Aldo Calcante  and Roberto Oberti

General comments

This research deals with the evaluation of several soil managements sytems in paddy-rice production in northern Italy. I think the topic is interesting and very suitable for the journal.

The article is now in a better shape than in the previous round.  I can see the authors have made an important effort to improve the manuscript and I'd like to congratulate them on it. Only a few small changes remain to get a more detailed work, some of which refer to formatting, and the rest to equations.

Introduction

The introduction has been greatly improved

Lines 45/6: more than, no more that

Line 51: crop, soil and climate

Line 71/2: in order to promote so that which can supplement … Is that right? It's a pretty strange sentence.

Material and Methods

Lines 143/5: “The cost per hectare ….. fungicide”. It is a result, and it should be placed in the proper place

Line 157: total work time. The authors wrote euro/ha, but total work time is measured in h. There is some mistake here.

Line 171. There's an extra parenthesis: (euro/year)) and an extra comma too (euro/h,)

Line 225: Do you mean because it is quite typical? I think “it is” must be added

I have some queries about cost calculation. Possibly the calculations are right and only modifications are required in the Materials and Methods section, but I cannot assure it. Please check these questions and and make modifications if necessary.

The method developed by ASABE has been explained. It is quite an important improvement. Nevertheless, some figures and formulae are still missing. For instance, table 3 shows hourly cost, and table 7 cost per area. I suppose the author used cost functions for the calculation. But this information is not provided (at least for one of the two). These functions should be added to the article, or, alternatively, the way used to get the hourly costs and surface costs. The equation 2 is wrong, regarding depreciation and interest cost. In the first case, authors have chosen a constant depreciation. Ok, but the right formula is: (initial value-final value (at the end of the life))/machine life. This is not what authors wrote. In the second case: 0.5*(initial value+final value at the end of the life of the machine)*interest rate. The third part of the equation is ok. Costs of repair and maintenance: regarding this kind of cost, the authors should consider that, the equation 4 developed by ASABE give us the total value of repair and maintenance throughout the life of the machine. In euros. Thus, the result has to be divided into the total number of working hours (estimated, of course, or extracted of ASABE tables, similar to table 3 of this study) in the life of the machinery.

Results and discussion

Line 301. At the end of the section, the authors write about total costs. This is ok, but from a general framework, a sentence should be added regarding future studies (for instance) since the yield should increase or decrease. I mean, the readers should be advised that conservation agriculture, in addition to reducing mechanization costs -as shown in this research- it also could decrease the yields and the economic balance is a priori uncertain for farmers.

Tables

Table 1. I suppose that the second part of the table is common to the three soil management systems. Please indicate it in a footnote. Title: delete the first word (The)

Table 2. Title: delete “The” too, as in the previous table. In general, the tables should be self-explanatory. Thus, please specify what Dr is in the footnote.

Table 3. The labour cost is missing for all the machines but the tractors. Those machines have also depreciation, repair, maintenance, and so on. Consider that the cost of any operation is equal to the sum of the cost of the tractor plus the cost of the machinery employed in that individual operation

Table 4.  At the beginning of the table, it can be read Minimum tillage and no-tillage, but conventional tillage was not written (it is missing). Please add it.

Figures

Figure 2. This figure is unnecessary, please delete it. The information of the part a) can be read in the table 5. The information of the part b) can be obtained without any effort  from the table 6 (we simply have to divide into 75).

Conclusions

This section is now ok

Author Response

See the attachment

This manuscript is a resubmission of an earlier submission. The following is a list of the peer review reports and author responses from that submission.

Round 1

Reviewer 1 Report

The paper analyze a classic theme in conservation agriculture, that it is very necessary. From my point of view, knowing the response of cropping systems to each geographical context is vital to understand them better. However, I think the work requires some changes and a deeper analysis. Introduction section is not satisfactory. It raises the state of the question and the objectives but not the causes that explain the need for this work, i.e., what is the authors contribute in this paper in relation to the benefits of the CA that the published papers have not already done?. Material and methods section is incomplete (see comments below). Results and discussion section is more a presentation of results than a discussion of them. It is necessary to interpret the results and broaden the discussion (see comments below).

Abstract

Lin. 20. ‘NT plots’ is a mistake, you must say ‘MT plots’.

Introduction

Lin. 33. Repeated ‘cover’ 3 times on the same line. The repetition of terms in the same sentence should be avoided.

Lin. 46-48. The citations of the studies referred are missing.

Lin. 49-52. This reflection requires a reference. Furthermore, I think it should go in the context of the lin. 37, about environmental benefits.

Materials and methods

Lin. 76-78. ‘Loam’ is not correct, it should be ‘silt’. The analysis of soil characteristics only by texture is insufficient. Furthermore, an average value is provided for the three plots. The analysis should be independent for each plot, and other basic soil properties and characteristics should be included, such as organic matter content, content of change bases and cation exchange capacity. I have not found in this section the method of calculating the texture. This section should be divided into subsections: (1) Study area, (2) Management and preparation of plots, (3) Methods of analysis.

Fig. 1. The meaning of CT, MT and NT must be indicated in the title.

Lin. 116. Here is 10 cm, but in the lin. 84 is indicated 20 cm.

Table 2. I think that a table is not necessary to indicate 3 values. They can be included in the text. Furthermore, the table indicates that it only applies to MT.

Lin. 140. The concept "effective field capacity" is important in the paper. For this reason it requires a more complete analysis: origin and background, how it is calculated, unit of measurement, etc.

Lin. 147. ‘agro-chemical’ is a generic concept that also includes fertilizers. It is convenient to be more explicit.

Table 3. The title appears twice ‘used’. The repetition of terms in the same sentence should be avoided.

Results and discussion

Table 5. I think that the dates of the operations should be included.

Tables 5 to 8. CT treatment must be indicated on the first line.

Lin. 214-216. The reference has only been cited, there is no discussion of results. Show comparative results and discuss them.

Lin. 217-228. The results are indicated but not discussed. There is no bibliographic analysis to discuss these results.

Fig. 1 (lin. 232) is really fig. 2. The quality of the graphics is not good.

Lin. 240-242 and 243-247. It is necessary to explain what are the causes of these results. It is not just about citing the paper, it is necessary to discuss the results.

Lin. 244. The word "profit" appears twice in the sentence.

Author Response

Sincerely

Aldo Calcante

Reviewer 2 Report

ARTICLE:  A technical-economic comparison between conventional tillage and conservative techniques in paddy-rice production practice in northern Italy.  Aldo Calcante  and Roberto Oberti

General comments

This research deals with the evaluation of several soil managements sytems in paddy-rice production in northern Italy. I think the topic is interesting and very suitable for the journal.

Unfortunately, it is clear for me that it needs a major improvement. Emphasis is needed in the different parts of the manuscript in order to get a more detailed, fine and comprehensive article.  In this sense, a lot of work still needs to be done by the authors. I hope we can see it in a new round. With the suggested changes, it would be a research of interest.

The methodology require an explanation on the way to calculate machinery costs. Currently this part is almost completely missing. This strong drawback should be radically improved. This is important because the methodology (and their assumptions) can influence the final results.

Likewise, a more in-depth discussion is needed, and in connection with the results of other authors.

I would like to see a new version of the manuscript incorporating the suggested changes.

Introduction

Lines 25/6: the primary goal of conservation agriculture (CA) is to cover soil in order to reduce soil loss (but not the reduction of production costs by itself). Regarding this issue, there are (or possibly there are) several additional benefits. One of these benefits could be or could not be cost reduction. Please change the focus of the line.

Line 33. Cover of cover?

Line 37. This is not true: reduction of rainfall? CA does not reduce rainfall at all, nor does it increase it.

Lines 37/43: At the beginning there are several lines to define and frame this type of agriculture, including its potential benefits. It is important to point out that conservation agriculture provides a wide variety of soil ecosystems services. Authors can complete this point including or claryfing another benefits that CA imply in the soil. As an example Carbonell-Bojollo et al (2012) explored sequestration potential of different types of residues (http://dx.doi.org/ 10.5424/sjar/2012103-562-11), Rodriguez-Lizana et al (2017) reported a multi-year study with a permanent organic cover in a wide area (https://doi.org/10.1002/ldr.2734), and Rodríguez-Lizana et al (2019) reported a multi-year study regarding soil protection with several types of cover (http://dx.doi.org/10.1016/j.agee.2018.05.012).

Line 40. Is expected.

Line 41. Reduced draw? Is that right?

Line 43/4: of the intensity.

Line 46. “some other research” … Ok, but what research is this? Please also explain the main causes for such conclusions. Why did those authors conclude that minimum tillage was more expensive than conventional tillage regarding production costs?

Line 56. Hallowed? I suppose you mean allowed.

Line 62 and following. This is interesting. A drop in yield of abount 20% could be devastating for many farmers. What did happen? Another climate?

In the introduction section, the author indicate and cite several studies with opposing results. This enables us to know that sometimes CA is benefitial, and other times it is not. Could the author establish a general framework or to cite several causes for such results? In principle, it could be somewhat confusing for the readers of Agronomy.

Material and Methods

Line 83. Please change the sentence. Is there an extra “was”?

Line 85/6. Which was the sowing rate? The same applies to line 87. Dates and rate are missing. It is also missing in table 1. More emphasis is needed to clarify this agronomic part.

Line 90/1. “Each plot ….investigated”. Delete, this is not necessary

Line 105. 4WD. You mean four-wheel-drive. I suggest the authors to define the abbreviation for the readers.

In general ,sections 2.1, 2.2 and 2.3 should be very summarized (or even some information merged with the table 1 so as to avoid unnecessary replications). For instance, the sentence “Seeding was carried out around mid-May using a 122 combined seed drill with 4.5 m working width, at a seeding distribution rate of 200 kg/ha” can be read in lines 111/2 and 122/3. It is also specified in lines 130/1 (thus, 3 times). Another example: N application rate 75 kg/ha in lines 104, 115 and 126. Please merge and reduce the length of the paper.

Line 107. Define what the false seedbed technique is.

Lines 155/6. In my opinion, this is one of the main drawbacks of the paper. Consider that this manuscript is about a technical-economic comparison. Machinery is fundamental in cost estimation (as well as production factors). As far as I can see, the article focuses primarily on the latest. Machinery cost is missing: there are only 3-4 very general lines (159/162) in the paper. This strong drawback should be radically improved, taking into account that this is the part of methods. This is important because the methodology (and their assumptions) can influence the final results. Please define the types of costs and the way (including formulae, perhaps in a table) to calculate each.

Line 176. Euro/kg or euro/L?

Line 178. This farm … typical … Please rewrite

Lines 184/5: this is not useful, since a proper experimental design was not conducted. Additionally, you have only one year. This sentence should be deleted.

Results and discussion

Please indicate the formula for calculating the effective field capacity, because some readers may be unaware of it. Regarding this question, the formula is dependent on the field efficiency, but no figures are given in this regard. In this sense, more clarity is needed.

Also, the way to calculate some costs is not well specified. Thus, is it quite difficult to review this part of the work. I suggest the authors - for a next round of this article- to clarify the methods employed, and the figures used, so as to be comparable with published studies and with future research.

Lines 239/242. This paragraph is curious. In lines 184/185 the authors state that “only for illustration purposes ….”, but here there is no problem in asserting that yields of NT exhibit a drop of about of 20%. This way of proceeding is not the right one in research. Consider that no experimental design was conducted. No ANOVA is possible at all. No repetitions in the test. Therefore, such comparative statements cannot be made since the results are not comparable. The same happens with lines 243-247. This is even stated in the conclusions …..

In addition, the comparison of yields with only a one-year experiment has no sense. Several years are required.

At the global level, more in-depth discussion is needed, and in connection with the results of other authors.

Conclusions

I feel I cannot review the conclusions of this article since, as previously written, several parts of the research are not clear enough. Some parameters, or even costs, are not as understandable as they should be for Agronomy in their current state. Some affirmations are not true, as have been explained before.

Tables

Table 1. The dates are missing, and they are completely essential to define the crop cycle. The format is really difficult to read. For instance: 1st fertilization is divided into 4 parts!!: 1st+Fertil+isatio+n. How is this possible? This is not the proper way of presenting a research paper. With respect to the herbicide treatment I can read 2 kg/ha (Glyphosate). Is that rate or does the author refer to L/ha? In line 108 it is written L/ha. Finally, please define operation in the last row.

Table 2. This table shows 3 figures. A table is not needed for that.

Table 3. The format is not adequate, and it should be modified.

Table 4. How this you establish the economic parameters? This is important but the source has not been quoted. Does it come from your own knowledge, or from ASABE? Is the tractor driver's labour included or not? This is hard to follow in its present form

Tables 5/6. Why is CT missing?

Author Response

Sincerely

Aldo Calcante
